**communications** engineering

# Stress-induced artificial neuron spiking in diffusive memristors
D. P. Pattnaik [1] ✉, Y. Sharma [2], S. Savel'ev [1] ✉, P. Borisov [1], A. Akhter[1], A. Balanov [1] & P. Ferreira[2]

Diffusive memristors owing to their ability to produce current spiking when a constant or slowly changing voltage is applied are competitive candidates for development of artificial electronic neurons. These artificial neurons can be integrated into various prospective autonomous and robotic systems as sensors, e.g. ones implementing object grasping and classification. We report here Ag nanoparticle-based diffusive memristor prepared on a flexible polyethylene terephthalate substrate in which the electric spiking behaviour was induced by the electric voltage under an additional stimulus of external mechanical impact. By changing the magnitude and frequency of the mechanical impact, we are able to manipulate the spiking response of our artificial neuron. This functionality to control the spiking characteristics paves a pathway for the development of touch-perception sensors that can convert local pressure into electrical spikes for further processing in neural networks. We have proposed a mathematical model which captures the operation principle of the fabricated memristive sensors and qualitatively describes the measured spiking behaviour. Employing such flexible diffusive memristors that can directly translate tactile information into spikes, similar to force and pressure sensors, could offer substantial benefits for various applications in robotics.

Memristors as electrical circuit elements have only emerged recently[1]. Due to their potential to realise energy-efficient hardware for neuromorphic AI systems, memristive devices have received tremendous attention[2–4]. It has also been shown that memristors are capable to emulate certain biological neuron behaviour, which makes them promising for neuromorphic devices[5,6]. Various types of memristive device structures such as $Au/Ag/SiO_x/Au$[7,8], $Pt/Ag:SiO_x/Pt$[9–12], $TiN/HfO_x/AlO_x/Pt$[13], $Cu/ZnS/Pt$[14], $Ag/SrTiO_3/(La,Sr)MnO_3$[15], $Pt/NbO_x/Pt$[16] have been developed which utilizes the migration of metallic ions or oxygen vacancies resulting as the change in resistance between high resistance state (HRS) and low resistance state (LRS).

In this work, we focus on a subclass of so-called diffusive memristors with metallic Ag nanoparticles (NPs) embedded in insulating matrix of $SiO_2$[7–9,17]. For these memristive devices, the transition from HRS to LRS and vice versa occurs due to a field-induced diffusion of metallic NPs to coalesce together to form a conduction filament (CF) between the electrodes. Upon increasing the external voltage, a CF is completely formed between the electrodes above a threshold voltage ($V_t$) and is ruptured at voltage below the hold one ($V_h$). This type of volatile resistive switching has been utilized[9] to build artificial neurons, converting DC voltage to current spikes. The external voltage and temperature cause change of spiking regimes, interspike intervals, and other spiking characteristics as discussed in refs. 9,10.

Therefore, we expect that other external parameters such as mechanical stress or pressure can affect NP dynamics[18–20], and thus, artificial neuron response, making memristor-based electronic components to be good candidates for neuromorphic sensors.

Recently, there has been special interest in the development of flexible memristors[21–26]. However, the main focus has been on studies of the effect of substrate flexibility on neuromorphic devices endurance and reliability. Some papers have also discussed fabrication of artificial neural networks on flexible memristors for neuromorphic computation[23,27], but most of these technologies rely on applying an external electric voltage. As far as we know, the impact of external mechanical forces on the spiking behaviour of memristors has not yet been explored. Consequently, this research introduces a promising opportunity to use flexible volatile memristors as touch-perception sensors for various applications, including neural networks.

The application of decoding tactile information from sensors into electrical signals is intriguing and has a crucial potential for development of robotic systems and sensors. For example, various Field Programmable Gate Arrays (FGPA) based Spiking Neural Networks (SNN) have been employed to convert sensor data into spike patterns using artificial mechanoreceptors[28–32]. However, a major roadblock for the implementation of dense and efficient FGPA-SNN is the large power overhead. So there is a trade off between accuracy and energy efficiency[33]. Here, we report on

[1]Physics Department, Loughborough University, Loughborough, LE11 3TU, UK. [2]Wolfson School of Mechanical, Electrical, and Manufacturing Engineering, Loughborough University, Loughborough, LE11 3TU, UK. ✉e-mail: d.pattnaik@lboro.ac.uk; s.saveliev@lboro.ac.uk

**Fig. 1 | Fabricated diffusive memristor on PET (polyethylene terephthalate) substrate and IV characterization. a** A side view schematic of a diffusive memristor device, PET/Pt(30 nm)/Ag:SiO$_x$(100 nm)/Ag (5 nm), with an attached voltage source. **b** Typical I-V characteristics of a fabricated memristor measured in both positive (blue line) and negative (red line) voltage cycles. The average of the measured I-V is shown in black. The arrows indicate the direction of voltage change.

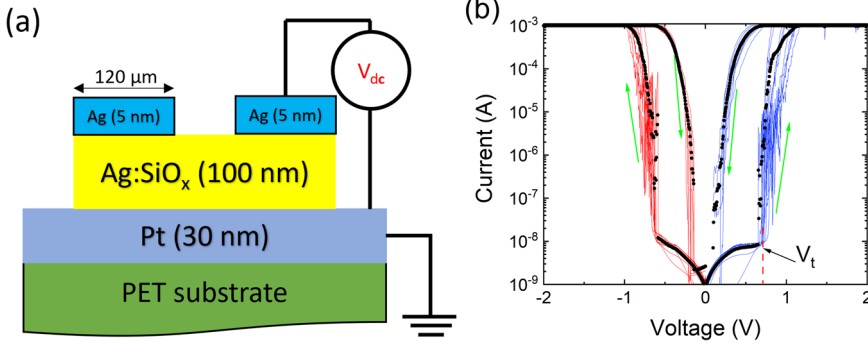

**Fig. 2 | Surface characterization of the diffusive memristor. a** X-ray photoelectron spectroscopy (XPS) survey of the fabricated diffusive memristor. **b–d** High resolution X-ray photoelectron spectra showing O 1s, Si 2p, and Ag 3d with the highlighted areas showing the spectra fits.

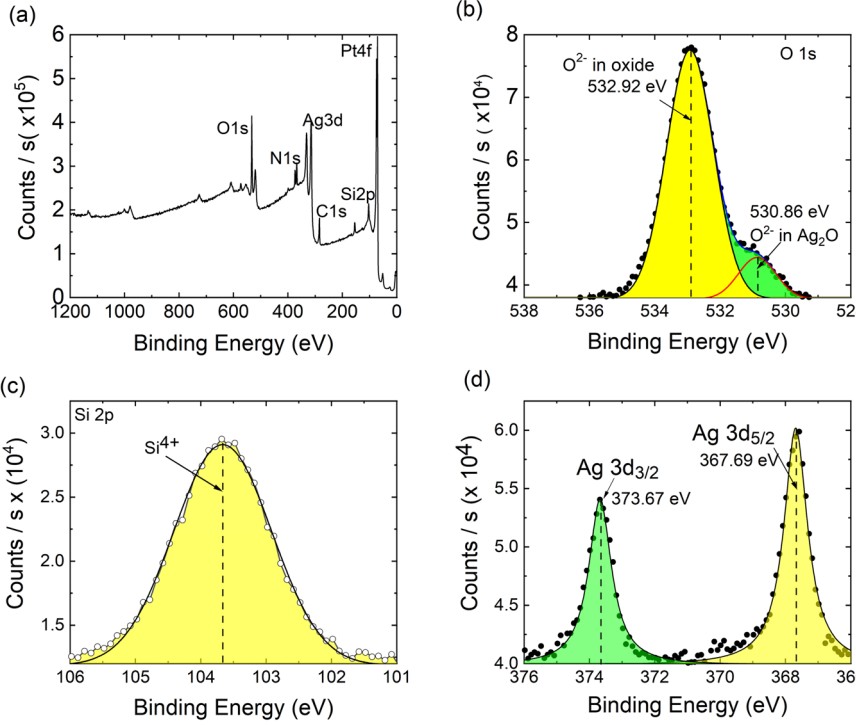

spiking characteristics of Ag NPs-based flexible diffusive memristors and show spiking dependence on the time interval between successive mechanical impacts and the corresponding pressure. The observed mechanical stimulation into electrical response is highly desirable for development of energy effective wearable and implantable electronics that can mimic e-skin function such as a mechanoreceptor sensor.

## Results and discussion
### Fabrication and characterization of flexible memristor
Diffusive memristors with structure Pt/Ag:SiO$_x$/Ag (see Fig. 1(a)) with 20% :80%Ag:Si atomic ratio were deposited on a polyethylene terephthalate (PET) substrate at room temperature using magnetron sputtering technique. A bottom Pt electrode layer, 30 *nm* thick, was deposited on a PET substrate. Following that, Ag and SiO$_2$ were co-sputtered in Ar to create a switching layer with a nominal thickness of *100* nm. Lastly, using a shadow mask with circles 120 *μ*m in diameter, 5 *nm* of Ag was sputtered to form the top electrodes.

The I-V loops of a device were recorded at room temperature employing a Keithley 4200 SCS analyzer equipped with an attached Everbeing probe stage. An external voltage was applied to the top electrode (see Fig. 1a) and varied within the range of -2 *V* to 2 *V* at a sweep rate of 50 *mV/s*.

The resulting I-V loops, shown in Fig. 1b, demonstrate clear volatile resistive switching behaviour for both positive and negative voltage sweeps. The observed I-V characteristics are nearly mirror-symmetric, with a transition from a high resistance state (HRS) to a low resistance state (LRS) occurring at a threshold voltage of $|V_t| = 0.65 \pm 0.05$ V.

To analyze the composition of the fabricated memristors, X-ray photoelectron spectroscopy (XPS) was performed. The XPS survey and the corresponding high-resolution spectra for oxygen O 1s, silicon Si 2p and silver Ag 3d are shown in Fig. 2 (a-d) respectively.

The XPS survey scan spectra is shown in Fig. 2a. Distinct peaks for Ag and Pt can be seen. The XPS spectrum for O 1s (Fig. 2b) shows two peaks[12,34] corresponding to silicon oxide (532.9 eV) and silver-oxide (530.8 eV) indicating partial oxidation of the silver NPs, while the remaining fraction of oxygen is contained in the dielectric silica matrix. This can also be attributed to oxidation of the top Ag layer which is even more likely given the surface sensitivity of the XPS. The Si 2p XPS spectrum (Fig. 2c) shows that silicon is in Si$^{4+}$ state and is part of silica. The Ag XPS 3d peak at 367.69 eV (Fig. 2d) corresponds to silver oxide (AgO)[12]. Therefore, we conclude that the Ag NPs are partially in metallic state and partially oxidized whilst being embedded in the dielectric matrix of SiO$_2$[9,11]. Based on our XPS data, we can state that the prepared memristors are similar and there is no degradation or

**Fig. 3 | Artificial memristive neuron. a** Electrical circuit of an artificial neuron with a diffusive memristor. **b** Measured memristor voltage spikes (red line) vs time for an external input voltage of 0.6 V (black line).

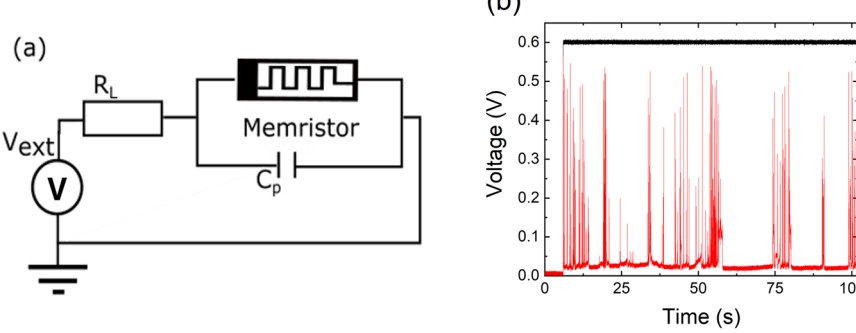

## Influence of mechanical impact on artificial neuron spiking

An artificial neuron device was constructed by connecting our diffusive memristor to an external resistor $R_L = 70$ kΩ and a capacitor $C_P = 1$ nF, as depicted in Fig. 3a. The values of $R_L$ and $C_P$ were selected based on previous studies of similar devices, e.g. in refs. 9,12, where electrical spiking behaviour was observed. When a constant DC voltage above a certain threshold was applied, for instance, $V_{ext} = 0.6$ V, the artificial neuron began generating voltage spikes across the memristor, as illustrated in Fig. 3b. This demonstrates a spiking behaviour similar to that previously reported for non-flexible diffusive memristors (see refs. 9,10). Note that relatively large capacitance of 1 nF in an integrated circuit could be potentially replaced with a lower capacitance and a correspondingly higher load resistance $R_L$ to maintain the same circuit time constant $R_L C$. As above previous studies showed, this parameter is crucial as it determines the generation and characteristics of the voltage spikes. However, this proposed approach requires further experimental verifications.

To investigate effects of mechanical stress on artificial neuron spiking behaviour for our flexible diffusive memristors, we have applied a direct mechanical impact on the memristor using an in-house built mechanical impact station as shown in Fig. 4a, b. A vertical mechanical impact up to 0.4 *MPa* with a minimal delay between two impacts of 0.2 *s* was applied to the top electrode using a pneumatically controlled stressor, whilst the memristor was covered with a 0.5 *cm* thick rubber layer and hosted on rubber covered platform[35]. For the experiments involving mechanical impacts, we slightly modified the configuration of the artificial neuron, cf. Fig. 4b and Fig. 3a. Specifically, as illustrated in Fig. 4b, the memristor was connected in series with an external capacitor $C$ and a parallel leakage resistor $R$, which improved the stability of the spiking measurements under repetitive impacts. The capacitance of $C_P$, previously used in the measurements without impact, illustrated in Fig. 3a, was replaced by the internal parasitic capacitance $C_i$ of the memristor originating from the electrodes and the active layer. It is reasonably to assume that $C_i$ changes as the device subjected to pressure. A constant voltage of 1.5 *V* was applied to the neuron whilst the voltage drop across the resistor $R$ was monitored for voltage spikes. We believe that the role of the external load and parallel capacitance as in Fig. 3a was fulfilled by the device contacts resistance combined with the external leakage resistance, and by the device internal capacitance as shown in Fig. 4b, respectively. The external capacitor in series was used to stabilise the spiking dynamics as well.

The time dependence of voltage spikes in response to impacts of 0.15 *MPa* (black line) and 0.35 MPa (red line) applied and then interrupted for 0.2 s, are shown in Fig. 4c. It must be noted that the device did not show any spiking behaviour for impact < 0.15 MPa.(See supplementary information Supplementary Note 1). When the memristor is spiking, two different regimes have been observed. For an impact of 0.15 *MPa* (Fig. 4c) the external capacitor voltage spikes occur from low to high values, which means the voltage across the memristor switches

from high to low values, that is the memristor staying most of the time at HRS occasionally transits to LRS for a short time (Fig. 4c (left black)). Such spiking behaviour eventually stops after a series of impacts [Fig. 4(c)(right black)] that is, the device remains permanently in LRS. However, for impact pressure of 0.35 *MPa*, the spiking behaviour gradually revives[Fig. 4c (left red)] after several impacts (~4200) and the spikes occur from top to bottom (LRS to HRS for the memristor)[Fig. 4c (right red)].

Further to this, the I-V loop characteristic was re-measured, and showed asymmetry (Fig. 4d) in comparison to a fresh sample with no impact history (Fig. 2b). The spiking frequency has reduced as well (Fig. 4e). An explanation for this is discussed in the next section. However, the ability to recover to HRS with an I-V characteristic demonstrates that the device has a memory.

## Artificial neuron spiking rates under repeated mechanical impact

We measured spiking rate at different time intervals between successive application of impact $t_p$ (Fig. 5). It is observed that the average spiking rate decreases for all impact pressure values, 0.2 *MPa* - 0.4 *MPa* as the time interval between successive impacts is increased. In this instance, we have implemented a fresh diffusive memristor layer without any prior impact history. This ensures that any potential impact damage (further discussed in next section and shown in Fig. 6(b) (d)) does not interfere with the spike rate analysis.

The change of the spiking rates under the influence of the mechanical impact can have two origins. The first one is due to internal capacitance change as a result of impact and deformation. Since the internal device capacitance is directly involved in the time constant of the artificial neuron, we believe this effect is responsible for the spiking rate change as discussed in the next section. Secondly, the additional pinning centres produced after repeated mechanical impacts are likely responsible for the transformation between two different spiking regimes as observed in Fig. 4c and explained in ref. 9, that is, whether the memristor remains most time in HRS and occasionally switches to LRS or vice versa.

In order to understand how mechanical impact influences the Ag NP distributions, we obtained SEM images of conductive clusters in the memristor before (Fig. 6c) and after (Fig. 6d) several impacts. The observed notable change of the size and distribution of Ag NPs allows us to propose physical mechanisms and mathematical model described in the next section.

## Model of artificial neuron spiking under repeated mechanical impact

When an external voltage ($V_{ext}$) is applied, the device internal capacitance ($C_i$) is charged until the voltage across the memristor reaches $V_t$, causing HRS to LRS switch. When in LRS, $C_i$ starts discharging until voltage drops to $V_h$, causing it to reset to HRS. A series of such charging and discharging processes under the influence of $V_{ext}$ generates a sequence of voltage spikes[9].

To explain the effect of mechanical impact on the spiking behaviour of the fabricated artificial neurons, we argue below that two different physical mechanisms can affect spiking and its intensity: (i) changing the time

**Fig. 4 | Stress-induced spiking. a** Experimental setup showing the pneumatically controlled impact station. The memristor is placed on the platform with a white rubber cover. The stressor is periodically pressed onto our memristor from the top. **b** Circuit diagram for the stress-induced spiking measurement. **c** Measured voltage spikes for impact pressure 0.15 MPa (black) and 0.35 MPa (red) with a 0.2 s time delay between successive impacts. The memrsitor spikes (left black) from high resistance state (HRS) to low resistance state (LRS) as a response to the impact which eventually stops (right black) as it is stuck in a permanent LRS. The spiking action is retrieved by gradually increasing the impact (left and right red) from 0.15 MPa to 0.35 MPa in steps of 0.05 MPa. **d** I-V characteristics of the device measured post several impacts in both positive (blue line) and negative (red line) voltages showing asymmetry in $V_t$. The curved arrows indicate the direction of the change in device resistance. **e** Measured memristor voltage spikes (red line) vs time for an external input voltage of 1.9 V (black line) for the device after ~4200 impacts.

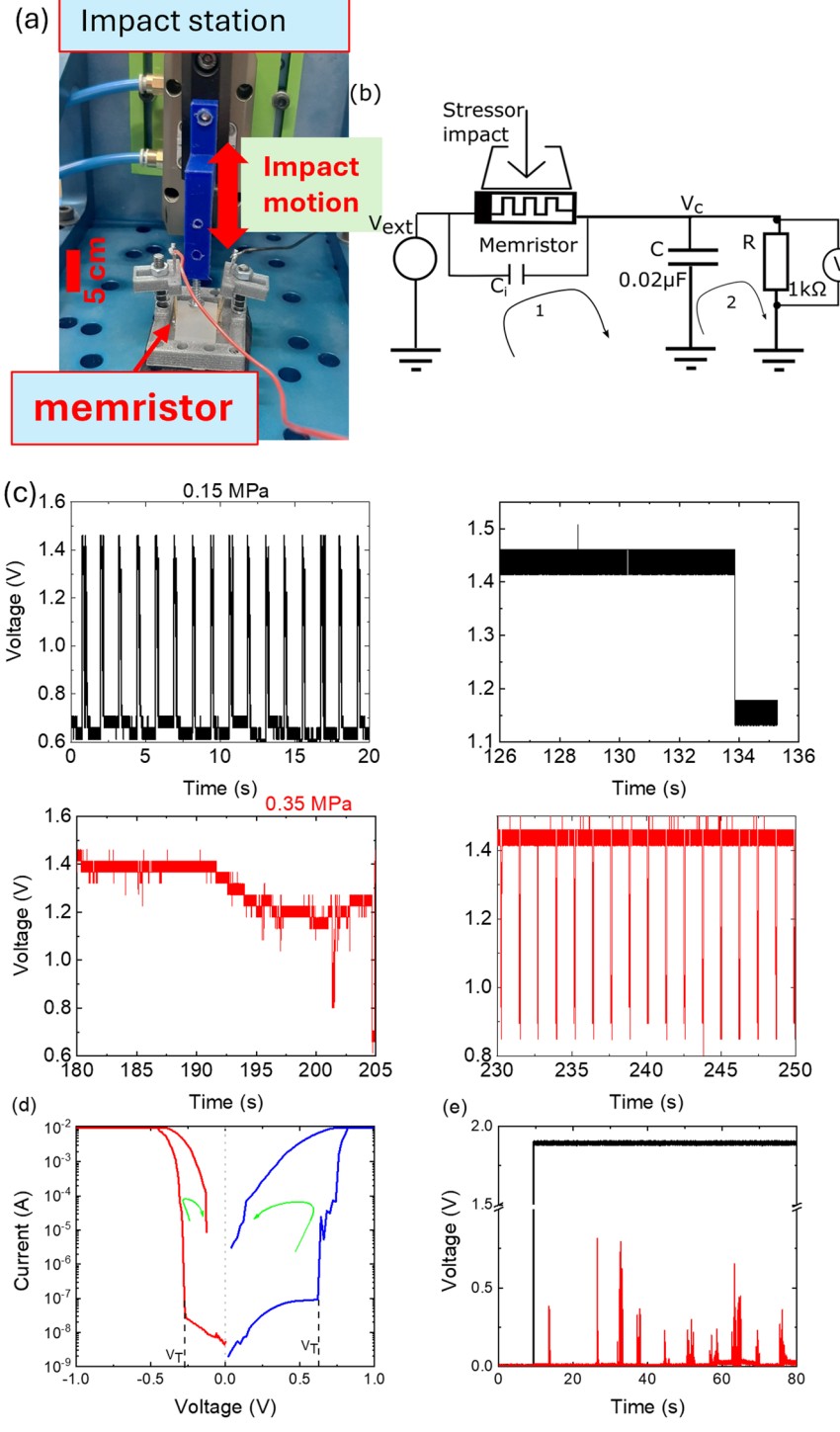

constant of the artificial neuron capacitor and (ii) varying the electrochemical profile where Ag clusters diffuse.

**Effects of pressure on the internal capacitance of a memristor.** When an external mechanical pressure is applied, the memristor undergoes deformation. The internal capacitance of our memristor ($C_i$) is:

$$C_i = \frac{A\epsilon}{d} \quad (1)$$

with an area $A$, the thickness $d$ between the electrodes and dielectric permittivity $\epsilon$. The relative change in the internal capacitance due to small

deformations can be estimated as:

$$\frac{\Delta C_i}{C_i} = \frac{\Delta\epsilon}{\epsilon} + \frac{\Delta A}{A} - \frac{\Delta d}{d} \quad (2)$$

The first term is the variation of the dielectric constant due to the impact, while the last two terms are the geometric changes to the studied memristor film.

The effect of deformation on the dielectric constant is called electrostriction enhancement of capacitance and is explained by the deformation-induced anisotropy of dielectric properties. A detailed explanation about this effect has been given in ref. 35 and ref. 36. While the contributions of $\Delta A$ and $\Delta\epsilon$ to $\Delta C_i$ are relatively small under the impact pressures used in the

experiment, (see Supplementary information (Supplementary Note 5) for estimation of dielectric constant alterations) the change in the effective distance between the electrodes due to pressure can be substantial. As shown in the comparison of Fig. 6b–d, the impact causes to form larger Ag clusters with characteristic sizes comparable to the value of $d = 100$ nm. The presence of large and thicker metallic inserts between the capacitor plates notably reduces the effective value of $d$, leading to a notable change in capacitance. Consequently, this change in capacitance affects the charging time of the capacitor and, therefore, the current interspike intervals, which is in agreement with our measurements.

Another contribution to change of capacitance originates from the modified dynamics of Ag particles. It can be argued that the deformation process leads to a reduction in the effective distance between the nanoparticles (NPs), facilitating the formation of a conductive filament (CF). Furthermore, the rearrangement of silver (Ag) NPs alters the local electrical field distribution, contributing to changes in the dielectric constant ($\epsilon$).

**Influence of pressure on the potential landscape for Ag clusters diffusion.** The SEM images presented in Fig. 6b, c illustrate the migration of Ag NPs toward the top electrode surface after repeated mechanical impacts (~4200 impacts) on individual devices. Consequently, an increased asymmetry in the current-voltage (IV) curves is observed following these impacts (Fig. 4d), indicating the generation of additional pinning centers for Ag NPs. Another contribution to the

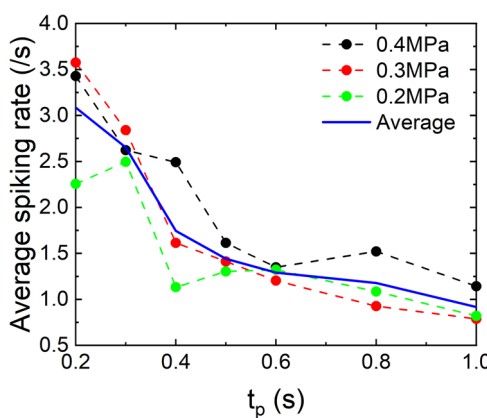

**Fig. 5 | Spiking rate.** Average spiking rate (number of spike per second) vs time interval between the impacts, at different impact pressures.

spiking rates changes originates from mechanical impact influencing the diffusion of Ag NPs in the deformed layer, see the schematic in Fig. 6b. This effect is similar to a change of the mobility of charge carriers in silicon under strain[37]. The influence of matrix lattice on a diffusion of conductive particles in memristors and modification of the electrochemical potential due to immobilised pinned charges, both of these effects can be influenced by pressure, and have been considered in ref. 38. All these physical mechanisms can be reduced to a modification of electrochemical potential of Ag NPs and can be evidenced by re-distribution of Ag NPs. Indeed, this can be another important contribution to the spiking rate changes under repeated mechanical impact. This phenomenon is further reflected in the measured voltage spikes, where less frequent spikes with lower spiking amplitude are observed for higher input voltage (Fig. 4e).

**Modelling artificial neuron spiking under mechanical impact.** To model the spiking rate changes in our artificial neuron in response to external mechanical impact, we consider interplay between three degrees of freedom: diffusion of Ag clusters forming conducting filaments (CFs) Eqn. (3a), Joule heat sinking to substrate Eqn. (3b), and electric current dynamics in the circuits described by Kirchhoff equations Eqn. (3c). To simplify the problem we consider the case when notable changes of the resistance occur due to only one Ag cluster sitting in the bottleneck of an almost formed CF[9,39]. Under this assumption, the set of equations (see Supplementary information (Supplementary Note 3 and 4)) describing an artificial neuron can be written as:

$$\eta \frac{dx}{dt} = -\left(1 + \Delta_U f(t)\right) \frac{\partial U}{\partial x} + q \frac{V}{L} + \sqrt{2k_B \eta T} \xi(t)$$
$$\frac{dT}{dt} = \frac{V^2}{C_h R(x)} - \kappa(T - T_0) \tag{3}$$
$$\tau \left(1 + \frac{C_2 - C_1}{C_1} f(t)\right) \frac{dV}{dt} = V_{ext} - \left(1 + \frac{R_{ext}}{R(x)}\right) V.$$

In this case the resistance depends only on the location $x$ of the bottleneck Ag cluster in the gap between almost formed parts of the filament, and can be approximated as[9] $R_0 = \cosh(x/\lambda)$ with the electron tunneling length $\lambda$ and all distances normalised by the gap size and the resistances by the minimal memristor resistance as described in the Supplementary information (Supplementary Note 3).

Accordingly, the impact of pressure can be reduced to two main factors: the modulation of capacitance due to changes in the dielectric constant and the effective distance $d$ between terminals, as well as the modulation of the height of the effective electrochemical potential $U(x)$. All these effects

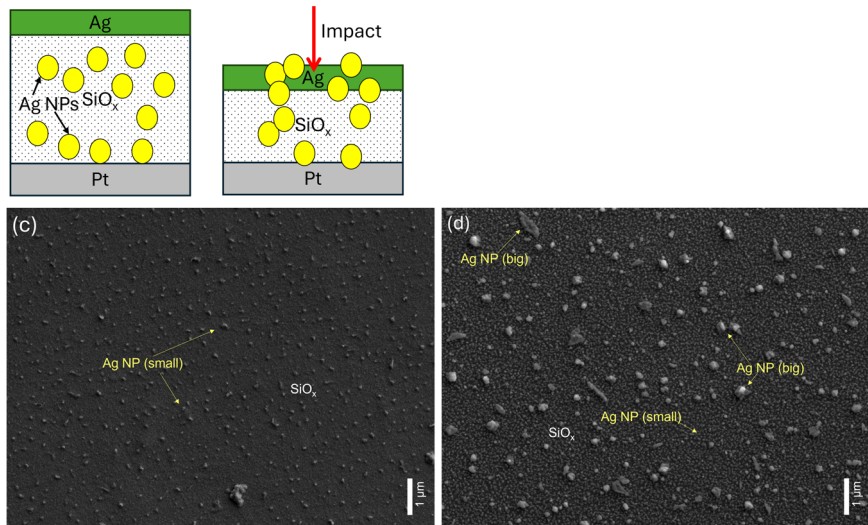

**Fig. 6 | Effect of impact on sample surface.**
**a** Schematic showing the migration of Ag NPs in the initial state and (**b**) due to impact induced deformation. **c, d** Scanning electron microscope (SEM) photograph showing Ag NPs on the sample surface before and after several mechanical impacts.

(a) Initial state (b) Deformed state

**Fig. 7 | (a) Simulated spiking rate. a** Average simulated spiking rate (number of spikes detected during time interval $\tau$ using criteria threshold $G = 0.05$) as a function of $t_p$ (idle time between impact application) when the pressure affects (**a**) only RC-time (i.e., $\Delta_U = 0$) with $(C_2 - C_1)/C_1 = 0.8$ and (**d**) only electrochemical potential (i.e., $C_1 = C_2$) and $\Delta_U = 0.4$. We simulated the dimensionless equations (see Supplementary materials where all simulation parameters are provided (Supplementary Note 3)). **b, c** and (**e, f**) Show spiking in form or relative memristor conductance vs time, and correspond to simulations (**a**) [**d**] for $t_{impact}/\tau = 0.05$ and 2, shown in (**b**) [**d**] by blue and red circle symbols.

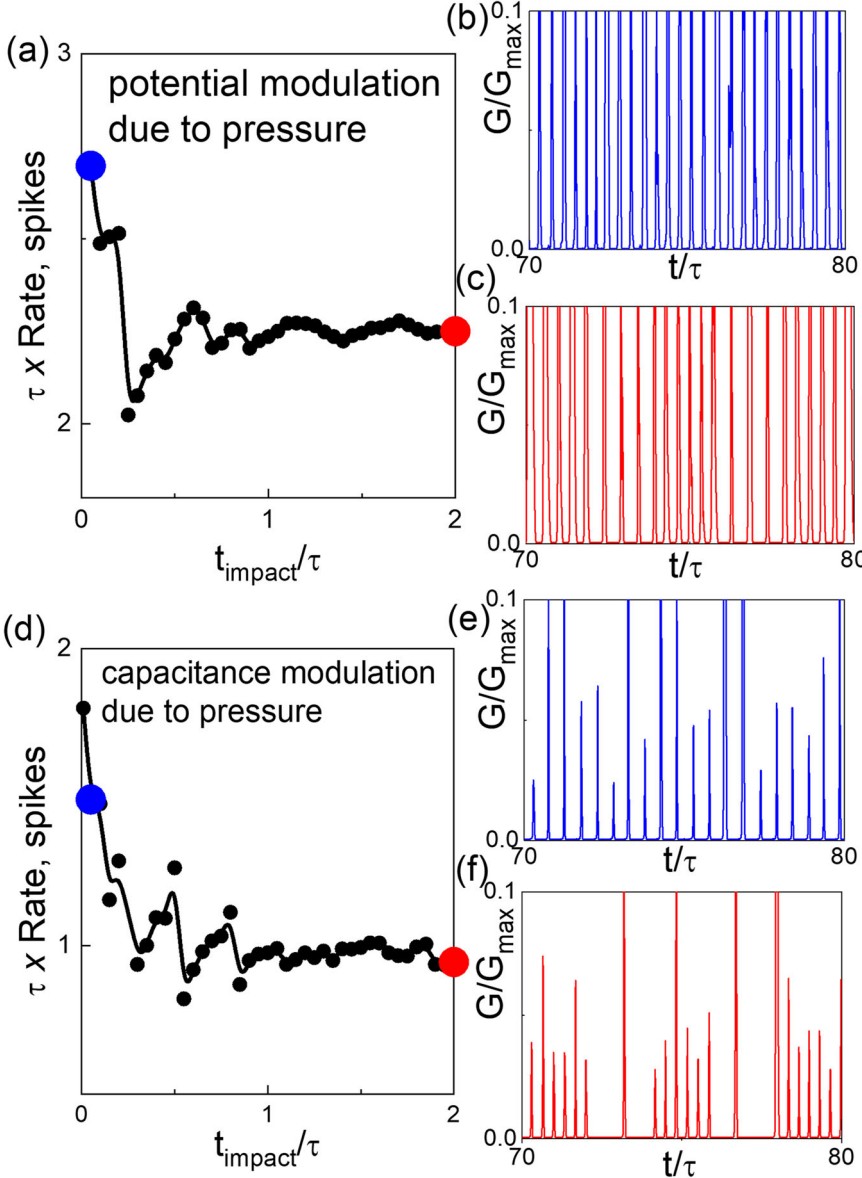

can be incorporated into the model by introducing explicit time dependence associated with the periodic (square wave) application of pressure. Specifically, we model the change in internal capacitance between $C_1$ and $C_2$, resulting in oscillations of the artificial neuron RC-time between two values $\tau = R_{ext}C_1$ and $R_{ext}C_2 = \frac{\tau C_2}{C_1}$. Assuming that such a change in the capacitance occurs abruptly with switching period $t_p$, we can approximate $f(t) = \text{sign}\left(\sin\left(\frac{2\pi t}{t_p}\right)\right)$ with $\text{sign}(y) = 0$ if $y > 0$ and $\text{sign}(y) = -1$ if $y < 0$. As a simple model of electrochemical profile oscillations, we study the square-wave modulations of potential height as follows: $U(x) \rightarrow (1 + \Delta_U(f(t)))U(x)$. In this approximation potential abruptly changes from $U(x)$ to $(1 + \Delta_U)U(x)$ with frequency $1/t_p$.

Our simulations as shown in Fig. 7 demonstrate that in both cases (modulation of capacitance only or modulation of electrochemical potential only), the rates can decrease with decreasing impact frequency. For modulation of potential height (See Fig. 7a), the interspike intervals are more noticeably affected, resulting in a rate decrease. For capacitance modulations (Fig. 7b), a decay of spiking rates with increasing $t_p$ occurs primarily because of the heights of the spikes decrease, causing some of them to fall below the detection threshold. This decrease is also accompanied by several "damped" oscillations, similar to those observed in our experiments.

## Conclusion

In conclusion, we have fabricated diffusive memristors with silicon oxide and silver NPs on a flexible PET substrate. XPS analysis along with SEM microscopy confirms the incorporation of metallic Ag NPs into the SiO$_x$ dielectric matrix. Current spikes under constant electrical voltage and repeated mechanical impact showed that there is a strong dependence of current spikes on the impact pressure and frequency. We have observed that the spiking can abruptly stop after several impacts due to a permanent conductive state. The high-resistive state can be retrieved by increasing the pressure of the impact. We have also demonstrated the dependence of average spiking rate of our memristor on the time interval between successive impacts. These experimental finding were explained and reproduced by a numerical model that involves variation of the internal memristor capacitance or electrochemical potential upon repeated mechanical impact. The obtained dependence of the spiking rate on impact idle time $t_p$, at different pressures, in addition to the memory of its I-V characteristics opens a possibility to utilize this device as a pressure sensor which is able to estimate the touching strength and also perform in-material computation (e.g., reservoir computing) by utilizing its fading memory of mechanical deformation history.

The use of such memristors that can directly decode tactile information into spikes, similar to force sensors or pressure sensors, could have outstanding benefits in various robotics and in-material computing applications[40–44], such as artificial prosthetic hands and grasping in robotics. This could enable the robots and autonomous systems to obtain biological like perception without requiring external bridges or complex processing, potentially enhancing their ability to interact with the environment in a more human-like manner.

## Methods

### Sample preparation

The studied diffusive memristors were deposited on a Polyethylene terephthalate (PET) substrate at room temperature by using magnetron sputtering technique. A bottom Pt electrode of 30 nm was deposited on the substrate at 50 W dc power. The diffusive memristor layer of nominal thickness 100 nm was made by co-sputtering from Ag and $SiO_2$ targets at 20 W and 300 W respectively in 0.85 Pa Ar pressure. A thin 5 nm Ag reservoir layer was deposited which also acts as the top electrode. A rectangular shadow mask was used during the deposition to gain access to the bottom Pt electrode.

### Electrical characterization

I-V characterization for the fabricated memristor was made using a Keithley 4200 SCS and an Everbeing probe station with 2 $\mu m$ tungsten tips. For the artificial neuron spiking measurement, a voltage pulse (0.6 V, 100 s) was applied to the device using a Rigol waveform generator and the device voltage was recorded using a PicoScope digital oscilloscope, whilst the memristor was connected in series to a load resistance $R_L = 70 \, k\Omega$ and in parallel to a capacitor $C_P = 1$ nF.

The measurements for the effect of impact on the memristor were performed using the in-house developed impact station discussed in the main text. For this measurement, the same memristor was put on the impact station test bench, with a rubber on top. The memristor was connected to an external RC block with $R = 1 \, k\Omega$ and $C = 0.02 \, \mu F$.

### Surface characterization

X-ray photoelectron spectroscopy (XPS) was performed using a Thermo K-Alpha system with an Al K$\alpha$ mono-chromated (1486.6 eV) source with an overall energy resolution of 350 meV. The analysis area captured was ~100 $\mu m$ x 200 $\mu m$. A survey scan was first performed to preview the chemical composition, subsequent high-resolution scans were then performed on the elements of interest before fitting their peaks to identify elemental state. All scans were charge corrected to adventitious C 1 s (C-C, C-H) peak at 284.8 $eV$. Scanning electron microscopy (SEM) was carried out on a JEOL 7100F. Imaging was done at both 5 $kV$ and 2 $kV$ to achieve the best resolution and surface detail of small particles.

## Data availability

The authors declare that the data supporting the findings of this study are available within the paper.

## Code availability

The authors declare that the simulation code supporting the findings of this study are available within the paper.

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

## Acknowledgements

D.P. thanks Alan Turing Institute (London) for the postdoctoral endowment fund (2022 Cohort), which was essential to develop the experimental methodology. The authors would like to thank Dr. Sam Davis for the XPS characterization and SEM imaging and acknowledge the use of the facilities within the Loughborough Materials Characterisation Centre (LMCC). This work was supported by The Engineering and Physical Sciences Research Council (EPSRC), grant no. EP/S032843/1.

## Author contributions

DP conceived the concept, prepared the samples, performed electrical and surface characterisation and wrote the first draft of the manuscript. DP,PB,AB,SS wrote the manuscript.YS,AA,PF designed the experiment and analyzed the measured data led by DP and PB. SS,AB performed the numerical simulations. All authors contributed to the discussion and polishing the manuscript.

## Competing interests

The authors declare no competing interests.
