## [Peer Review File · Communications Engineering]

Stress-induced artificial neuron spiking in diffusive memristors

Corresponding Author: Dr Debi Pattnaik

Version 0:

Reviewer comments:

Reviewer #1

(Remarks to the Author)

In the present manuscript, the authors disclose a study on stress-induced oscillatory behavior in volatile switching CBRAMs. The approach is quite interesting. There are, however, several open questions and missing information as stated below. Thus, the manuscript requires major revisions and re-evaluation.

- 1) The size of the top electrode is not given.
- 2) What is the sweep rate used in the experiments.
- 3) Physical quantities should appear in italics.
- 4) In Fig. 3 and the corresponding experiment it does not become clear to which electrode the voltage is applied.
- 5) The authors mention that the memristor is connected to a parallel capacitor of 1 nF. This is quite a high value and requires a rather large capacitor. How can this be implemented in an integrated circuit?
- 6) The authors mention a stress of 0.15 MPa to 0.35 MPa under which the oscillatory behavior is observed. How does this range compare to the intended application, i.e., tactile sensors.
- 7) The authors mention that no spiking behavior is observed for a stress less than 0.15 MPa. For clarification, the authors should also show that data.
- 8) On page 7, line 150 ff, the authors mention that the change in dielectric constant is consistent with the observed experimental trends. Please add real values to prove the consistency.
- 9) The authors report that the I-V curves shift after repeated impacts. How many impacts are required? How does this behavior influence/limit the performance of the spiking circuit?
- 10) The model used by the authors is only briefly explained. More information is necessary. Moreover, the parameters set for the simulation (including the peripheral elements) are not provided. Thus, it is impossible to reproduce such simulations. All necessary information and parameter values need to be given.
- 11) In Fig. 6, the authors only compare the general trends of simulation and experiment. While for both the spike rate decreases with the delay between impacts, the absolute values are completely off. Please explain the difference.

Reviewer #3

(Remarks to the Author)

This paper reports on the effect of mechanical impacts on the spiking response of artificial neurons made from Ag nanoparticle-based diffusive memristors. This work represents an important task for the future development of artificial neurons capable of interacting with their environment, and could find interesting perspectives in the field of robotics and autonomous systems. However, the experimental data presented in the current form of the paper show clear inconsistencies that's why I recommend a major revision before publication to clarify the following points:

- Could the authors justify the choice of RL and Cp values selected to design the oscillator circuit presented in figure 3a?
- On figure 3b, the measured memristor neuron voltage exhibits stochastic spiking whereas tonic spiking (stable periodic spiking) would be more suitable to study the impact of mechanical stress on the neuron spike rate. How could the circuit parameters be tuned to get stable and periodic spiking?
- It is not clear why the authors have used a different oscillator circuit than in figure 3 to study the stress-induced spiking in figure 4. Could the author justify this technical choice?
- From the first oscillogram in figure 4c, the measured voltage spikes seem to be stable and periodic with a period of around 1.2s but the time delay between two successive impact is of only 0.2s. How can the spike interval be higher than the mechanical impact interval? Why the response of the neuron is not stochastic anymore?
- Still in figure 4c, the memristor neuron shows a degradation of its spiking characteristic under mechanical cycling (i.e. the spiking behavior stops after a series of impacts). This degradation is attributed to the migration of Ag NPs within the stack as revealed by SEM analysis in figure 5. Could the authors propose any technological solution to avoid this reliability issue?

Could the authors also better justify why spikes are again observed with the same rate of 0.8/s when the impact pressure is increased up to 0.35MPa and why the spike sign is reversed?

- Several parameters of equation 3 are not defined in the paper. Could the authors define all the parameters used in the model?

- There is a manifest inconsistency between figure 6a and figure 4c. From figure 4c, an average spike rate of around 0.8/s can be extracted whereas an average spike rate of 18/s is presented in figure 6a for an interval between successive impacts of 0.2s. Could the authors clarify this?

Version 1:

Reviewer comments:

Reviewer #1

(Remarks to the Author)

The authors answered all comments raised convincingly. I recommend accepting the manuscript.

Reviewer #3

(Remarks to the Author)

The authors have responded convincingly to the various questions/remarks raised by the reviewers. They have also corrected and modified their article accordingly. For these reasons, I recommend publication of the article in its current form.

Reviewers' comments:

Reviewer #1 (Remarks to the Author):

In the present manuscript, the authors disclose a study on stress-induced oscillatory behaviour in volatile switching CBRAMs. The approach is quite interesting. There are, however, several open questions and missing information as stated below. Thus, the manuscript requires major revisions and re-evaluation.

Response: We thank the reviewer for the generally positive review our manuscript. In the revised version, we addressed the raised questions and added missing information, also improved our manuscript for targeting a broader readership. The changes in the manuscript are highlighted in blue.

1) The size of the top electrode is not given.

We thank the reviewer for raising this. The size of the top electrode is 120 μm in diameter. We have added this information in the main draft.

Lastly, using a shadow mask with circles 120 μm in diameter, 5 nm of Ag was sputtered to form the top electrodes.

2) What is the sweep rate used in the experiments.

We thank the reviewer for asking this. The sweep rate is 50 mV/s. We have added the following statement in the main draft.

External voltage was applied to both the top and bottom electrodes (see Fig. 1 (a)) within the range of -2V to 2V, utilizing a sweep rate of 50 mV/s.

3) Physical quantities should appear in italics.

We thank the reviewer for pointing this out. We have now changed the font type of the physical quantities.

4) In Fig. 3 and the corresponding experiment it does not become clear to which electrode the voltage is applied.

Thank you for your question. We have clarified this by changing the figure 1 (a). The new figure shows the top electrodes where the voltage is applied with a ground connection to the bottom electrode.

5) The authors mention that the memristor is connected to a parallel capacitor of 1 nF. This is quite a high value and requires a rather large capacitor. How can this be implemented in an integrated circuit?

Ans- We thank the reviewer for raising this question. The work described in this manuscript is a proof of concept for the effectiveness of the diffusive memristor as a tactile sensor. The engineering task of integrating a 1 nF capacitor in parallel is an open-ended question.

However, in order to produce spikes, the RC time is the quantity we would like to control; that is, the load resistance should also be considered. As discussed previously, the capacitance can have lower values if the load resistance is increased correspondingly. We have added the discussion with references in the main article to reflect the suggestion:

6) The authors mention a stress of 0.15 MPa to 0.35 MPa under which the oscillatory behavior is observed. How does this range compare to the intended application, i.e., tactile sensors.

Ans. We thank the reviewers for this discussion.

Commercial tactile sensors, such as Force Sensing Resistors (FSRs), usually operate within a voltage range of 0 to 5 volts. We have now repeated the same experiment by replacing our memristor with a commercial FSR sensor (Interlink electronics FSR 408). For impact pressure 0.15 MPa and 0.35 MPa, the voltage output is shown below:

It is observed that for different impact pressure, a commercial sensor responds by a higher voltage readout. The sensor only responds by changing the voltage amplitude.

We have shown this in the supplementary information, see Fig. S2.

As can be seen, the sensor output are triangular waveforms, which depend on the mechanical impact only. However, in our case we have observed a time and impact dependence of voltage spikes which can be incorporated into spiking neural networks.

7) The authors mention that no spiking behaviour is observed for a stress less than 0.15 MPa. For clarification, the authors should also show that data.

Ans: We thank the reviewer for a clarification. We have added the data for the pressure below 0.15 MPa in the supplementary section.

8) On page 7, line 150 ff, the authors mention that the change in dielectric constant is consistent with the observed experimental trends. Please add real values to prove the consistency.

We thank the reviewer for this question. We have made estimations of the expected effect on dielectric constant and the relative change in the dielectric constant at 0.35 MPa was 0.1%, see our supplementary information. Based on that, we amended our discussion and obtained new simulation results for a modified model, see Figs 7a and b. Our model assumes now the impact modifies the potential for the particle transfer, see discussion in the main draft. We believe such model is more realistic than the originally proposed dielectric constant variation.

9) The authors report that the I-V curves shift after repeated impacts. How many impacts are required? How does this behaviour influence/limit the performance of the spiking circuit?

Thank you for this important question.

A set number of impacts was not measured as the same device was used for several values of pressure and for long periods of time. However, the measurements were approximately performed for 2 minutes each at different pressure and time delays. A rough calculation with average time delay 0.2 sec would nominally mean an approximate 600 impacts for each pressure. The number of pressure impacts from 0.1 mPa to 0.4 mPa in steps of 0.5 mPa, is equal to 4200.

The effect of impact towards the IV and spiking characteristic has now been shown in Fig. 4 (d) and (e)

We have edited this paragraph to reflect the changes.

10) The model used by the authors is only briefly explained. More information is necessary. Moreover, the parameters set for the simulation (including the peripheral elements) are not provided. Thus, it is impossible to reproduce such simulations. All necessary information and parameter values need to be given.

Thank you for the comment. In the revised draft, we have added a supplementary file to include the dimensionless version of the equations together with the parameters used in simulations:

We have also explained the model in more details in the main draft to accurately predict the dynamics of the artificial neuron under impact.

11) In Fig. 6, the authors only compare the general trends of simulation and experiment. While for both the spike rate decreases with the delay between impacts, the absolute values are completely off. Please explain the difference.

Thank you for asking this question. The reason of this apparent discrepancy is due to the different time scales. In simulations, the time is normalised by $R_{ext}C/30$. For these reasons we intentionally decided not to compare absolute values in the simulation and experiment.

Reviewer #3 (Remarks to the Author):

This paper reports on the effect of mechanical impacts on the spiking response of artificial neurons made from Ag nanoparticle-based diffusive memristors. This work represents an important task for the future development of artificial neurons capable of interacting with their environment, and could find interesting perspectives in the field of robotics and autonomous systems. However, the experimental data presented in the current form of the paper show clear inconsistencies that's why I recommend a major revision before publication to clarify the following points:

We thank the reviewer for the positive feedback on the work. We also welcome the questions and have answered them below. We hope that the revised version makes the manuscript more appealing to wider readers.

1) Could the authors justify the choice of RL and Cp values selected to design the oscillator circuit presented in figure 3a?

We thank the reviewer for this question. The values of RL and Cp were chosen following our previous investigations in similar devices (e.g., Physical Review Applied 19, 024065 (2023)) where we induced electrical spiking and our intentions to set the device near the threshold voltage of spiking. In this point the sensitivity of our system to external actions are high, enabling an efficient transfer of impact stress to electrical spiking.

We have added the reference and the reasoning to use these parameters in the main text (shown in blue)

2) On figure 3b, the measured memristor neuron voltage exhibits stochastic spiking whereas tonic spiking (stable periodic spiking) would be more suitable to study the impact of mechanical stress on the neuron spike rate. How could the circuit parameters be tuned to get stable and periodic spiking?

We completely agree that our device demonstrates stochastic spiking, and more deterministic spiking will be more suitable to study the impact of mechanical stress on artificial neurons. In order to get more stable and periodic spiking, the following engineering controls can be implemented at the device level.

- a) Fabricating devices with much a smaller switching layer area. By this, when biased a single conduction channel would be formed instead of multiple ones. A smaller structure will also help to have stable threshold and hold voltages.
- b) Fabricating electrical gratings/ nano channel guides- This will be used to channel the nanoparticles to only grow and break in a single direction.

In addition, we have demonstrated that diffusive memristors can have different spiking modes with different degrees of stochasticity Physical Review Applied 19, 024065 (2023), Nanoscale 15, 15665 (2023), Chaos, Solitons & Fractals 145, 110803 (2021), thus the more regular spiking can be potentially achieved by varying external resistance, capacitance and applied voltage. However, such fine tuning is outside of the scope of this paper,

3) It is not clear why the authors have used a different oscillator circuit than in figure 3 to study the stress-induced spiking in figure 4. Could the author justify this technical choice?

We thank the reviewer for asking this question regarding difference in measurement circuit scheme.

We felt the stress-induced spikes were too stochastic, so we add a capacitor in series to modify the overall time constant and to stabilise the spiking pattern. The corresponding discussion is provided in the main draft and highlighted in blue.

4) From the first oscillogram in figure 4c, the measured voltage spikes seem to be stable and periodic with a period of around 1.2s but the time delay between two successive impact is of only 0.2s. How can the spike interval be higher than the mechanical impact interval? Why the response of the neuron is not stochastic anymore?

Thank you for the question.

As we explained earlier, the artificial neuron demonstrates self-sustained oscillations due to different bifurcation mechanisms (see, e.g., Chaos, Solitons & Fractals 149, 110997 (2021)). When a high frequency impact is applied, the system starts to average the influence of pressure, which can drive the system to the self-sustained spiking regime. If the system is near the boundary of self-sustained spiking, the interspike intervals are usually very long and can be longer than the time between the successive impacts. As explained before, the degree of stochasticity depends on spiking modes: for example, as we describe in Chaos, Solitons & Fractals 149, 110997 (2021), the interspike intervals can be very long near bifurcation to a periodic spiking regimes.

5) Still in figure 4c, the memristor neuron shows a degradation of its spiking characteristic under mechanical cycling (i.e. the spiking behavior stops after a series of impacts). This degradation is attributed to the migration of Ag NPs within the stack as revealed by SEM analysis in figure 5. Could the authors propose any technological solution to avoid this reliability issue? Could the authors also better justify why spikes are again observed with the same rate of 0.8/s when the impact pressure is increased up to 0.35MPa and why the spike sign is reversed?

We thank the reviewer for this question.

As demonstrated in Physical Review Applied 19, 024065 (2023), the spiking may originate from different stages characterized by varying resistance levels, where the Ag-clusters predominantly reside. This leads to the observed change in the polarity of spiking, as highlighted by the reviewer. Based on our experimental findings, we infer that initially, Ag-clusters transition from high-resistance states to low-resistance states, where they remain stationary (as detailed in Physical Review Applied 19, 024065 (2023)). As the impact interval approaches 0.8/s, the system is likely to undergo another bifurcation, prompting Ag-clusters to switch back to high-resistance states. As previously demonstrated (Physical Review Applied 19, 024065 (2023)), increasing temperature or applying negative voltages can often facilitate resetting the system within operational parameters.

A proposed technical solution to avoid the issue with device endurance, other than changing the sample temperature and gamma irradiation, is by developing/ fabricating devices with much a smaller switching layer area. By this, when biased a single conduction channel would be formed instead of multiple ones. A smaller structure will also help to have stable threshold and hold voltages.

6) Several parameters of equation 3 are not defined in the paper. Could the authors define all the parameters used in the model?

Thank you for bringing this into consideration. We have added a supplementary file which now provides much more detailed description of the model and supply the model potential, and the phenomenological potential used in the simulations.

7) There is a manifest inconsistency between figure 6a and figure 4c. From figure 4c, an average spike rate of around 0.8/s can be extracted whereas an average spike rate of 18/s is presented in figure 6a for an interval between successive impacts of 0.2s. Could the authors clarify this?

We thank the reviewers for this question and pointed out the error with the spike rate.

Since the previous studied sample went through several impact cycles, we believe this may have changed the nanoparticle density. This needs to be studied in further details.

However, for the scope of this specific study, we remeasured using a different sample from the same batch, which had no previous impacts. We have updated the average spike rate in Figure 6 and have added the following lines in the main text.

In this instance, we have implemented a fresh diffusive memristor layer without any prior impact history. This ensures that any potential impact damage as shown in Fig. 4 does not interfere with the spike rate analysis